# Hybrid Tabu-Grey wolf optimizer algorithm for enhancing fresh cold-chain logistics distribution

**Hao Zhang, Jianing Yan, Liling Wang**[ID]*

School of Business, Beijing Technology and Business University, Beijing, China

* wliling1005@163.com

**Data Availability Statement:** All relevant data are within the manuscript and its Supporting information files.

## Abstract

The increasing public demand for fresh products has catalyzed the requirement for cold chain logistics distribution systems. However, challenges such as temperature control and delivery delays have led a significant product loss and increased costs. To improve the current situation, a novel approach to optimize cold chain logistics distribution for fresh products will be presented in the paper, utilizing a hybrid Tabu-Grey wolf optimizer (TGWO) algorithm. The proposed hybrid approach combines Tabu search (TS) and Grey wolf optimizer (GWO), employing TS for exploration and GWO for exploitation, aiming to minimize distribution costs in total and establish efficient vehicle scheduling schemes considering various constraints. The effectiveness of the TGWO algorithm is demonstrated through experiments and case studies compared to other heuristic algorithms. Comparative analysis against traditional optimization methods, including Particle swarm optimization (PSO), Whale optimization algorithm (WOA), and original GWO, highlights its superior efficiency and solution quality. This study contributes theories by demonstrating the efficacy of hybrid optimization techniques in complex supply chain networks and dynamic market environments. The practical implication lies in the implementation of TGWO to bolster distribution efficiency, cost reduction, and product quality maintenance throughout the logistics process, offering valuable insights for operational and strategic improvements by decision-makers. However, the study has limitations in generalizability and assumptions, suggesting future research areas including exploring new search operators, applying additional parameters, and using the algorithm in diverse real-life scenarios to improve its effectiveness and applicability.

## 1. Introduction

With the improvement of people's living standards and the change in consumer attitudes, there is a growing demand for fresh products, which is effectively driving consumption and creating new growth opportunities for the fresh e-commerce industry. However, the perishable nature of fresh food means that it needs to be kept fresh for a certain period of time, otherwise, the quality and taste will be impacted, and it even poses a risk to human health. This is where cold chain logistics distribution comes in, which refers to the use of professional

**Funding:** This paper was supported by Beijing Natural Science Foundation (9222007).

**Competing interests:** The authors declare that there are no Competitive financial benefits and no conflict of interest.

refrigeration equipment and technical means in the logistics process to maintain suitable temperature and humidity for fresh food throughout the distribution process. Despite its importance, the cold chain logistics distribution process faces many challenges due to the unique nature of fresh food, such as unstable temperature control, cargo damage and delayed delivery time. These problems not only impact the quality and taste of fresh food, but also increase logistics costs and delivery risks. Shockingly, despite 15% of the world's total energy consumption already being dedicated to cold chain logistics and cooling systems, the annual loss of fresh products ranges from 5%-15% of the total production due to inadequate infrastructure and inefficient resource allocation [1]. In developing countries, the rate of loss can even reach up to 50% [2]. Furthermore, the expensive costs involved in the food cold chain limit the capacity for fresh products e-commerce to make a profit [3]. Therefore, logistics distribution optimization is necessary for the development of a fresh cold chain.

In the cold chain logistics distribution network for fresh products in cities, the distribution requirements (primarily temperature) vary depending on the type of product being transported. Additionally, each customer has a designated distribution time, which may differ from other customers' times. Some customers are flexible with their availability and may accept compensation for late deliveries, while others have strict time windows and will reject products delivered outside their allotted slot. This creates a unique challenge in the form of a Vehicle Routing Problem with Time Windows (VRPTW), where each customer has their own soft and hard time window, which was first proposed by Dantzig and Ramser [4]. Therefore, efficient vehicle schedules play a critical role in optimizing the old chain logistics distribution, with extensive scholars identifying the challenge, trying to improve this situation, and applying various methodologies, including mixed integer programme models and evolutional algorithms [5]. Followed by Amorim and Almada-Lobo's [6] theory of cost-freshness trade-offs, which will be explained in detail in Section 2. Although it can be seen that both soft and hard time windows exist in fresh cold chain logistics distribution. The key challenge that needs to be tackled is how to ensure that each customer is reached within their specified time window to their satisfaction.

Motivated by optimizing cold chain logistics distribution of fresh e-commerce, a mathematical optimization model is applied in this paper, aiming at optimizing cold chain logistics distribution for fresh e-commerce. The objective is to lower supply chain costs and ensure on-time delivery by optimizing food delivery routes while considering constraints, such as vehicle capacity, maximum travel range, minimum acceptable freshness level of food, and coexisting soft and hard time windows. In addition, a hybrid Tabu-Grey Wolf Optimizer (TGWO) algorithm was designed to solve the above problem, in which Tabu Search (TS) was used to increase the local search ability of the Grey Wolf Optimizer (GWO) and improve the quality of the initial solution by further exploring the search space. The results of the experiments proved the validity and potential application of TGWO in providing superior results compared to other common heuristic algorithms.

The remaining sections of this paper are organized as follows. The related literature will be reviewed in Section 2. In Section 3, the research methodology and solution approach will be presented, followed by Section 5, in which an overview of the results obtained will be provided. In the end, discussion and research limitations will be concluded.

## 2. Literature review

Many scholars have made significant efforts to enhance the optimization of vehicle routing of the fresh cold chain. To design delivery routes, some focused on the Vehicle Routing Problems (VRP), which was first introduced in 1959 by Dantzig and Ramser under the title

"The Truck Dispatching Problem" [4]. The study of the VRP and its extensions have given rise to major developments in the field of exact algorithms and heuristics [7]. Wang [5] proposed a mixed integer programming model and developed a branch-and-price algorithm to solve vehicle routing problems with drones. Amorim and Almada-Lobo [6] examined the relation between distribution scenarios and the cost-freshness trade-off. A∈ constrained method is used for small-size instances for VRPTW, and for large-size instances a multi-objective evolutionary algorithm is implemented. Wang [8] constructed a green and low-carbon cold-chain logistics distribution route optimization model and Cycle Evolutionary Genetics Algorithm (CEGA) was proposed to contribute to the model. Davila [9] used tabu search, chaotic search and general algebraic modeling to solve the vehicle routing problem for distributing refrigerated products. The VRPTW commonly employs two types of time window constraints: hard and soft. In the cold chain logistics distribution problem of fresh products, hard time window constraints require fresh products to be delivered during the exact designated time window, forbidding both earliness and tardiness [10]. Nonetheless, in reality, fresh products remain edible even if the hard constraint is violated, and the distribution network may become over-constrained, resulting in feasible solutions [11, 12]. Soft time window constraints handle the earliness and tardiness with penalties, which are often assumed to follow a linear relationship with the amount of variation from the time window [13]. Therefore, numerous exceptional works have been developed by combining VRPTW with various constraints and objectives to solve more practical instances in cold chain logistics. This paper mainly considers the constraints of vehicle capacity, maximum travel range, minimum acceptable freshness level of food and coexisting soft and hard time windows, aiming to optimize the food delivery routes to minimize total costs and ensure on-time delivery. As for solution method, instead of exact algorithm, more and more metaheuristic algorithms are proven to be suitable in cold chain distribution.

Currently, swarm intelligence-based algorithms are commonly used for optimizing cold chain logistics distribution, like ant colony optimization (ACO) [14–16], particle swarm optimization (PSO) [17–19], whale optimization algorithm (WOA) [20, 21], and grey wolf optimizer (GWO) [22, 23]. GWO was proposed by Mirjalili [24] in 2014. It simulates the hunting behaviour and social hierarchy of grey wolves in nature. The existing studies have shown that GWO is competitive with some common optimization algorithms [24, 25]. Because of the advantage of fewer control parameters, a highly stochastic nature and derivation-free mechanism, numerous variants of GWO have been proposed to solve complex constrained practical optimization problems [26, 27], for example, optimizing the power dispatch problem [28], the assembly flow shop scheduling problem [25], the capital goods scheduling problem [29] and the path planning problem [30].

However, as search space dimensions grow, GWO tends to perform poorly in exploitation [31]. Scholars have proposed various approaches to improve the original GWO algorithm. For example, to increase the diversity of the grey wolf population, Zhu [32] combined the GWO algorithm with the differential evolution (DE) algorithm to improve its performance and global exploration capability. Singh and Hachimi [33] balanced the developmental and explorational abilities of GWO by introducing the spiral equation of the whale optimization algorithm (WOA). Zhang [34] combined the GWO algorithm with the biogeographic optimization (BA) algorithm based on antagonistic learning, which can prevent the algorithm from quickly choosing the local optimal solution. Gupta and Deep [35] modified algorithm RW-GWO based on random walk and its performance is exhibited in comparison with GWO and state of art algorithms GSA, CS, BBO and SOS on IEEE CEC 2014 benchmark problems. The results demonstrated that the algorithm provides a better leadership to search a prey by grey wolves.

All these existing literature lays a theoretical foundation for this paper. GWO can effectively avoid local optimal solutions but its optimization accuracy needs to be improved, while tabu search algorithm has strong global search capability but is prone to local optimal solutions. To address these limitations, this paper proposes a hybrid Tabu-Grey wolf optimizer (TGWO) to enhance the optimization model as an extension of VRPTW. The proposed algorithm combines the exploration capability of the GWO algorithm with the local search capability of the Tabu search algorithm to achieve a better balance between exploration and exploitation. The specific enlightenment can be reflected in the following aspects: Firstly, as for shortcomings of the GWO algorithm in generating the initial solution, the tabu search algorithm with strong global search ability is introduced to expand the types of solutions and enhance the optimization ability of the algorithm. Secondly, although GWO algorithms are rarely used in similar discrete optimization problems, some scholars have improved algorithms to apply in vehicle routing problems and achieved great success. Therefore, this paper analyzed these improvement ideas, combined with the characteristics of the improved grey wolf optimization algorithm, proposed a solution method, and applied it to the VRPTW of fresh e-commerce cold chain distribution. Finally, based on the characteristics of fresh e-commerce cold chain distribution, when constructing the VRP model, the paper considered the coexistence of soft and hard time windows, and established a model with the maximum vehicle capacity and the maximum travel range constraints. The results show that it can provide better results compared to traditional methods and highly improve the efficiency and effectiveness of the cold chain logistics distribution process.

## 3. Methodology

### 3.1 Model assumption

To develop the mathematical model, the assumptions are summarized as: 1) The central depot has sufficient commodity supplies, and only one type of capacitated distribution vehicle is considered; 2) Each customer's demand must be satisfied, and each customer is only visited once by a distribution vehicle; 3) Each vehicle departs from and returns to the same central depot; 4) The distribution vehicles travel at a constant speed, which means that the transportation time is only associated with the accumulated transport mileage; 5) The distribution vehicles are only responsible for delivering commodities and temporarily preserving the rejected goods, if any, until returning to the central depot. They are not responsible for receiving new freight; 6) The cargo damage rate $\partial_2$ when unloading is higher than $\partial_1$ when in transport.

### 3.2 Sets, parameters, and variables

The parameters used in the model are shown in Table 1.

### 3.3 Cost analysis

The mathematical model aims to minimize the total cost, which encompasses the sum of the fixed cost, transportation cost of distribution vehicles, cargo damage cost, and penalty for violating time window constraints.

Fixed cost. The total fixed cost of the distribution vehicles includes equipment depreciation, potential losses during the driving process and the salaries for vehicle's crew. Besides, the fixed cost of a distribution vehicle is related only to whether the vehicle participates in the delivery

**Table 1. Sets, parameters and variables.**

| Sets | |
|---|---|
| $V$ | Central depot ($i = 0$) and the set of customers ($i, j = 1, \ldots, n$), $V = \{0, 1, \ldots, n\}$ |
| $V'$ | Set of customers with soft time windows |
| $V''$ | Set of customers with hard time windows |
| $K$ | Set of distribution vehicles |
| **Parameters** | |
| $c$ | Unit product price |
| $c_f$ | Fixed cost of distribution vehicle |
| $v$ | Travel speed of distribution vehicle |
| $D$ | Maximum travel range of distribution vehicle |
| $Q$ | Capacity of distribution vehicle |
| $Q_i$ | The cargo capacity of the vehicle when it leaves customer $i$ |
| $p_1$ | Oil consumption cost for the unit travel range |
| $p_2$ | Refrigeration cost for the unit transport time |
| $d_{ij}$ | Distance between customer $i$ and $j$ |
| $q_i$ | The demand of customer $i$ |
| $[ET_i, EL_i]$ | Time window in which customer $i$ expects to be served |
| $st_i$ | Service time in which customer $i$ is served |
| $et_i$ | Soft time window penalty paid when the vehicle arrives early to customer $i$ |
| $el_i$ | Soft time window penalty paid when the vehicle arrives late to customer $i$ |
| $et_i'$ | Hard time window penalty paid when the vehicle arrives early to customer $i$ |
| $t_{si}^k$ | Service time of vehicle $k$ to customer $i$, that is, unloading time at customer $i$ |
| $T_{ijk}^t$ | Time period of vehicle $k$ from $i$ to $j$ at time $t$ |
| $\partial_1$ | The cargo damage rate when the door remains closed during transportation |
| $\partial_2$ | The cargo damage rate when the door remains open during unloading |
| $M$ | An infinite number |
| **Variables** | |
| $x_{ij}^k$ | Equal to 1 if the delivery route between customer $i$ and $j$ is visited by vehicle $k$ and 0 otherwise |
| $y_i^k$ | Equal to 1 if vehicle $k$ provides delivery service for customer $i$ and 0 otherwise |
| $s_k$ | Equal to 1 if vehicle $k$ is currently being arranged to provide service and 0 otherwise |
| $t_{ik}$ | Time when vehicle $k$ arrives to customer $i$ |

service. The total fixed cost $C_1$ can be calculated as follows:

$$C_1 = \sum_{k \in K} c_f \cdot s_k \tag{1}$$

Transportation cost. The total transportation cost of the delivery vehicles includes oil consumption and refrigeration costs during the whole transportation process. It increases with the transport distance and time. Normally, the total transportation cost accounts for 20%~25% of the total operation cost in fresh distribution. In this paper, the total transportation cost $C_2$ can be calculated as follows:

$$C_2 = \sum_{k \in K} \sum_{i=0}^{n} \sum_{j=0}^{n} p_1 \cdot x_{ij}^k \cdot d_{ij} + \sum_{k \in K} \sum_{i=0}^{n} p_2 \cdot t_{ik} \tag{2}$$

Cargo damage cost. The quality of perishable products during cold chain logistics distribution is a crucial factor affecting customer satisfaction and repeat purchase in e-commerce again [36]. This study assumes that the temperature inside the transportation compartment

remains stable during transit, resulting in a constant spoilage rate of the perishable products. However, when unloading the goods, the opening door causes heat exchange with the environment, leading to a higher spoilage rate than during transit. As a result, the total cost of cargo damage comprises two parts: the cost incurred during transportation due to time extension and the cost of damage when unloading the goods at the customer's location.

Cargo loss during transportation:

$$E_1 = \sum_{k=1}^{K} \sum_{i=0}^{n} q_i y_i^k (1 - e^{-\partial_1 \sum_{i=0}^{n} \sum_{j=0}^{n} T_{ijk}^t z_{ik}}) \tag{3}$$

Cargo loss during unloading:

$$E_2 = \sum_{k=1}^{K} \sum_{i=0}^{n} Q_i y_i^k (1 - e^{-\partial_2 t_{si}^k}) \tag{4}$$

The cargo damage cost is determined by the cargo loss and the unit price of the delivery product, and the total cost of cargo damage during the distribution process is:

$$C_3 = c(E_1 + E_2) = \sum_{k=1}^{K} \sum_{i=0}^{n} c y_i^k \left[ q_i \left( 1 - e^{-\partial_1 \sum_{i=0}^{n} \sum_{j=0}^{n} T_{ijk}^t z_{ik}} \right) + Q_i (1 - e^{-\partial_2 t_{si}^k}) \right] \tag{5}$$

Penalty cost. When there is a soft time window constraint, the penalty cost can be calculated as follows:

$$penalty_1 = max \{et_i(ET_i - t_{ik}), 0, el_i[t_{ik} - (EL_i - st_i)]\} \tag{6}$$

When there is a hard time window constraint, the penalty cost can be calculated as follows:

$$penalty_2 = max\{et_i'(ET_i - t_{ik}), 0, M[t_{ik} - (EL_i - st_i)]\} \tag{7}$$

The penalty cost of the hard time window may be very large if vehicles arrive after the designated time; therefore, this paper moves the hard time window into the constraints. Then, the total penalty cost $C_4$ in the vehicle routing problem can be calculated as follows:

$$C_4 = \sum_{k \in K} \sum_{i \in V'} max\{et_i(ET_i - t_{ik}), 0, el_i[t_{ik} - (EL_i - st_i)]\} \tag{8}$$

### 3.4 Objective function

Based on the analysis above, the cold chain logistics distribution mathematical model of fresh products is presented as follows:

$$Min\, Z = C_1 + C_2 + C_3 + C_4 \tag{9}$$

The constraints of the model are as follows:

$$\sum_{j=1}^{n} x_{0j}^k = \sum_{j=1}^{n} x_{j0}^k \leq 1, \forall k \in K \tag{10}$$

$$\sum_{i=0}^{n} y_i^k \cdot q_i \leq Q, \ \forall k \in K \tag{11}$$

$$\sum_{i=0}^{n} \sum_{j=0}^{n} x_{ij}^k \cdot d_{ij} \leq D, \ \forall k \in K \tag{12}$$

$$\sum_{k \in K} y_i^k = 1, \ \forall i \in V \tag{13}$$

$$t_{ik} + st_i + \frac{d_{ij}}{v} - M\left(1 - x_{ij}^k\right) \le t_{jk}, \ \forall i, j \in V, \forall k \in K \tag{14}$$

$$t_{ik} \ge ET_i, \ i \in V'', \ \forall k \in K \tag{15}$$

$$t_{ik} + st_i \le EL_i, \ i \in V'', \ \forall k \in K \tag{16}$$

$$et_i \ge 0, el_i \ge 0, et_i' \ge 0, \ \forall i \in V' \tag{17}$$

Eq (9) is the objective function, which minimizes the total cost $Z$ in the vehicle routing problem. Constraint (10) indicates that distribution vehicles start from the distribution centre and return to the centre after finishing the deliveries to the given customers. Constraint (11) indicates that the demand of all customers in each delivery route should not exceed the maximum vehicle capacity. Constraint (12) indicates that the summation of the distribution distance in each delivery route should not exceed the maximum transport range. Constraint (13) ensures that each customer is only accessible by one delivery vehicle. Constraint (14) ensures the continuity of the delivery services. Constraints (15) and (16) ensure that vehicles start and finish the deliveries to customers within the hard time windows as required. Constraint (17) indicates that the penalty coefficient is a nonnegative number.

## 3.5 Conversion of TSP to VRP

In order to propose the solution idea, this paper introduces the concept of "Traveling Salesman Problem (TSP)", a special form of VRP, which given a set of cities and a cost to travel from one city to another, seeks to identify the tour that will allow a salesman to visit each city only once, starting and ending in the same city, at the minimum cost [37]. Although TSP is a simple version of VRP, it has long been proven to be an NP-Hard problem in combinatorial optimization.

Considering that the TGWO studied in this paper is an algorithm commonly used to solve continuous function optimization problems, and the settings of the upper and lower bounds of variables are limited when applied to discrete optimization problems such as VRP, this paper uses the following solutions: (1) to define each grey wolf as a TSP sequence starting from the distribution centre; (2) to define the total cost of the cold chain logistics as the fitness function. As in Fig 1, (b) traverses through each TSP sequence in ascending order of their indices, and inserts the starting point (distribution centre) if necessary, which involves assigning the rest of the distribution tasks to a new driver if the previous route violates any constraints

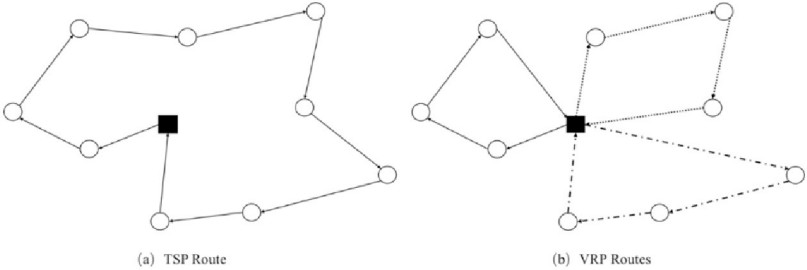

(a) TSP Route          (b) VRP Routes

**Fig 1. TSP route converted to VRP routes.**

(capacity, transport mileage or food freshness). After converting a set of TSP routes into one feasible solution to the proposed model, the fitness value is calculated for further comparison with other candidate solutions; (3) to select define this as the optimal solution in the current schemes by comparing the current solutions with the candidates.

## 4. Solution approach

### 4.1 Grey wolf optimizer (GWO)

The inspiration for GWO originates from the social hierarchy and hunting behaviour of gray wolves. There are four types of society members in a gray wolf pack, namely, alpha ($\alpha$), beta ($\beta$), delta ($\delta$) and omega ($\omega$), according to the descending order of their social hierarchy. Each wolf performs its own duties and cooperates with the other wolves. Accordingly, the best solution obtained by GWO is considered the alpha, the second and third solutions are called the beta and delta, respectively, and the remaining solutions are referred to as the omegas, which are also assumed to be inferior solutions. The principal steps are as follows.

**Step 1: Initialization**

Eq (18) presents the mathematical model of the initialization process:

$$x_{ij} = lower_j + \left( upper_j - lower_j \right) \times random, i = 1, 2, \ldots, N, j = 1, 2, \ldots, D \quad (18)$$

where $x_{ij}$ is the $j$th element of the $i$th gray wolf's position vector; $upper_j$ and $lower_j$ represent the upper and lower bounds of the $j$th element, respectively; $N$ is the total number of wolves, which equals $n + 1$ (the total number of distribution centres and customers); $D$ is the dimension of the wolves; and $random$ is an independent variable that has a value between 0 and 1. Thus, the initial gray wolf population can be expressed as $X_i = (x_{i1}, x_{i2}, \ldots, x_{iD})$, where $i$ takes a value from 1 to $N$.

It should be noted that in the vehicle routing model of this paper, each TSP route should start at the distribution centre. Thus, the first elements of each gray wolf's position vector will be maintained as the smallest number of iterations by setting the upper and lower bounds of the first column to the same infinitesimal real number, $lb_1$, and the lower bounds of the other dimensions are set to be the same number $lb_2$ ($lb_2 > lb_1$). An example of the operation mode is shown in Fig 2. Therefore, the random numbers in the first dimension would always be the first numbers when the elements of each gray wolf's position vector are arranged in ascending order. Then, based on the decoding mechanism introduced in the next step, the obtained position vectors can be transformed into several sets of routes, which all start at the distribution centre, pass through all the customer locations and remain at the location of the last customer served.

| | 1 | 2 | 3 | 4 | 5 | 6 |
|---|---|---|---|---|---|---|
| | (Distribution Center) | | | | | |
| Upper bound | 0.01 | 5 | 5 | 5 | 5 | 5 |
| Lower bound | 0.01 | 0.2 | 0.2 | 0.2 | 0.2 | 0.2 |

**Fig 2. An example of setting lower and upper bounds.**

**Step 2**: **Encoding and Decoding**

After the grey wolf population is randomly initialized, the random numbers obtained in each dimension of each wolf are arranged in ascending order. Then, several sequences of integers arranged from 1 to $n + 1$ are obtained. The original customer code numbers corresponding to these ascending sequences are the candidate solutions of TSP.

**Step 3**: **Solution Generating**

Each non-holonomic TSP route needs to be transformed into one VRP route based on customer orders and several different constraints. The transformation is as follows.

First, the corresponding demands of the customers are summed up in sequence according to the obtained TSP route. Once the current vehicle load or distribution distance is about to or has already met the constraint of maximum capacity $Q$ or the maximum transport mileage $D$, adding one more unscheduled customer to this route will result in an overload or running out of fuel on the return to the centre. The algorithm terminates the process of adding new customers and inserts number 1 (code number of the distribution centre) into the TSP route immediately before the code number of the following unscheduled customer. This indicates that the delivery route of the first vehicle has been finished. Then, consider the next delivery route arrangement. Continue the previous steps until all given customers are arranged. Finally, a set of vehicle routing schemes is obtained, in which 1 can appear more than once each time for a new route, and each customer order can only occur once.

**Step 4**: **Fitness Function**

This paper sets the formula for the total costs as the fitness function. The smaller the total costs are, the smaller the fitness value:

$$fitness_i = Z = C_1 + C_2 + C_3 + C_4 \tag{19}$$

After each iteration, the gray wolf with the smallest fitness value is defined as the alpha, and the wolves with the second and third smallest fitness values are defined as the beta and delta, respectively. In this way, the first three optimal gray wolves and their position vectors are obtained, as well as the corresponding candidate VRP routes.

**Step 5**: **Update the positions of all gray wolves based on those of the first three optimal gray wolves, α, β and δ.**

This updating process can be summarized in three sub-steps:

1. Gray wolves surround their prey during the hunting process, and the distance between wolf and prey can be expressed by the following formula:

$$\overrightarrow{D} = |\vec{C} \cdot \overrightarrow{X_p}(t) - \overrightarrow{X}(t)| \tag{20}$$

where $\overrightarrow{X_p}(t)$ and $\vec{X}(t)$ are the position vectors of the prey and a randomly chosen gray wolf, respectively. In addition, $t$ is the current iteration. $\vec{C}$ is the coefficient vector and is calculated as follows:

$$\vec{C} = 2 \cdot \overrightarrow{r_1} \tag{21}$$

where $\overrightarrow{r_1}$ is a random vector that varies between 0 and 1.

2. The position of the gray wolf is updated and can be expressed as follows:

$$\vec{X}(t+1) = \overrightarrow{X_p}(t) - \vec{A} \cdot \vec{D} \tag{22}$$

where $\vec{A}$ is the coefficient vector and is calculated as follows:

$$\vec{A} = 2\vec{a} \cdot \overrightarrow{r_2} - \vec{a} \tag{23}$$

where $\vec{a}$ linearly decreases from 2 to 0 over the course of the iterations and $\overrightarrow{r_2}$ is a random vector that varies between 0 and 1, which is the same as $\overrightarrow{r_1}$.

3. After the gray wolves locate the position of the prey (Eqs (24) ∼ (29)), they attack it (Eq (30)). This process can be expressed as follows:

$$\overrightarrow{D_\alpha} = |\overrightarrow{C_1} \cdot \overrightarrow{X_\alpha}(t) - \vec{X}(t)| \tag{24}$$

$$\overrightarrow{D_\beta} = |\overrightarrow{C_2} \cdot \overrightarrow{X_\beta}(t) - \vec{X}(t)| \tag{25}$$

$$\overrightarrow{D_\delta} = |\overrightarrow{C_3} \cdot \overrightarrow{X_\delta}(t) - \vec{X}(t)| \tag{26}$$

$$\overrightarrow{X_1} = \overrightarrow{X_\alpha} - \overrightarrow{A_1} \cdot \overrightarrow{D_\alpha} \tag{27}$$

$$\overrightarrow{X_2} = \overrightarrow{X_\beta} - \overrightarrow{A_2} \cdot \overrightarrow{D_\beta} \tag{28}$$

$$\overrightarrow{X_3} = \overrightarrow{X_\delta} - \overrightarrow{A_3} \cdot \overrightarrow{D_\delta} \tag{29}$$

$$\overrightarrow{X}(t+1) = \frac{\overrightarrow{X_1} + \overrightarrow{X_2} + \overrightarrow{X_3}}{3} \tag{30}$$

**Step 6: Termination Criterion**

Repeat the above iterative search process. Terminate the algorithm when the pre-set maximum iteration is reached, and then, output the optimal solution obtained so far as well as the corresponding delivery vehicle route and fitness value (total delivery costs). With the use of the mentioned algorithms, the solution to the vehicle routing problem can be determined.

The GWO procedure is illustrated in Fig 3.

## 4.2 Tabu-grey wolf optimization algorithm

GWO tends to provide poor convergence performance with increasing search space dimensions. Thus, it is necessary to enhance the local search ability of the GWO algorithm. Therefore, the neighbourhood searching process of TS is introduced into TGWO to strengthen its exploitation ability, and the tabu list limits the solution returned to recently visited solutions in a given move and increases the diversity of the candidate solutions to vehicle routing optimization problems.

The main steps of the TGWO algorithm are as follows.

**Steps 1–4 are the same as those of GWO.**

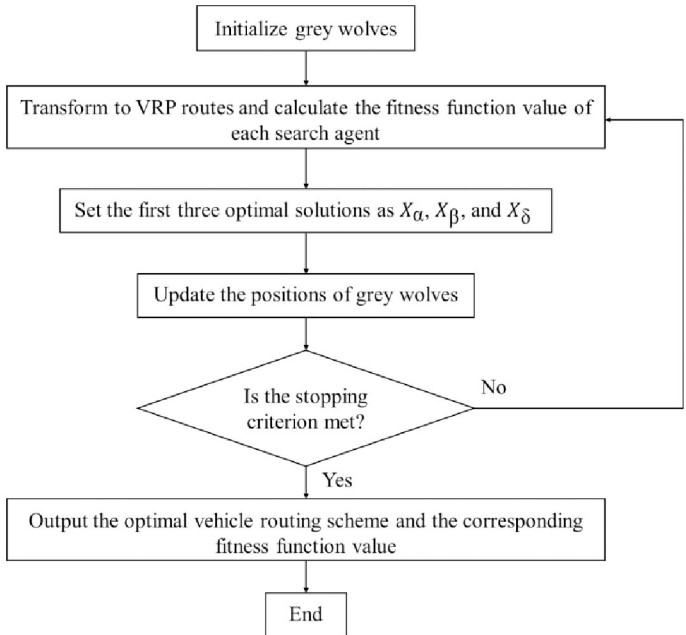

**Fig 3. The GWO procedure.**

**Step 5: Neighbourhood Search**

This is part of the section where TGWO differs from the standard GWO. Once a new feasible solution is generated in the iterative process, the fitness value needs to be calculated and compared with that of the best solution obtained so far. If the fitness value shows superiority, then the optimal solution should be replaced by the current one and a neighbourhood transformation should be performed to determine whether there is a better solution in the neighbourhood. This searching process is performed by running the neighbourhood searching process $Iter_{neighbor}$ times. $N_{neighbor}$ initial neighbourhood transformed solutions would be generated each time. Otherwise, the iterative process should be continued. The neighbourhood transformation further explores the search space and contributes to creating more feasible TSP routes and a better VRP solution. It improves the quality of the current solution.

A multi-neighbourhood structure is designed for TGWO, where three search operators are randomly adopted: insert, swap and 2-opt. This paper discusses the neighbourhood strategies in details by simulating a delivery route with seven customers: 1→3→5→4→6→2→7.

**4.2.1 Insert operator.** A customer is randomly selected and placed (inserted) after another random position in the delivery route. The customers after that position move further. The operation procedure is presented in Fig 4: a selected customer (framed by a dotted line) is inserted into a randomly selected position (marked by a positive triangle). Then, the original route is transformed into 1→3→2→5→4→6→7.

**4.2.2 Swap operator.** Interchange the positions of two randomly chosen customers to change the order through which the delivery vehicles pass in the TSP scheme. The operation procedure is presented in Fig 5: a selected customer (framed by a dotted line) is randomly selected position, after swap transformation, the delivery order becomes 1→2→5→4→6→3 →7.

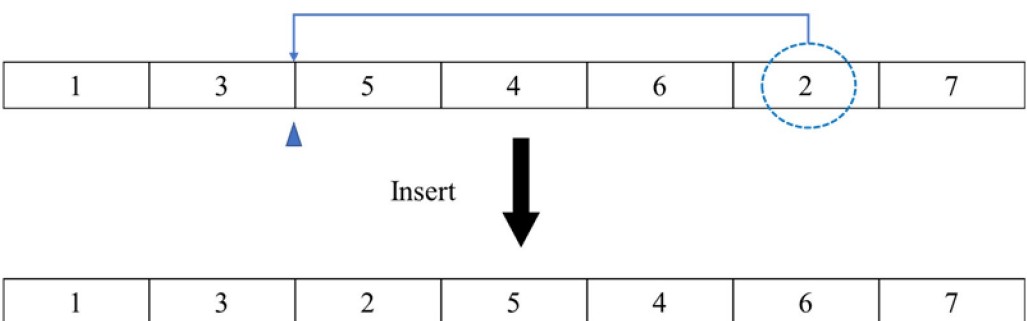

**Fig 4. An example of how to perform the insert operation.**

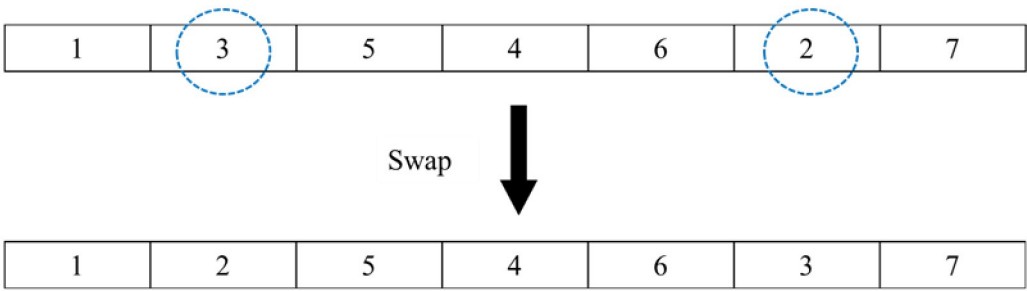

**Fig 5. An example of how to perform the swap operation.**

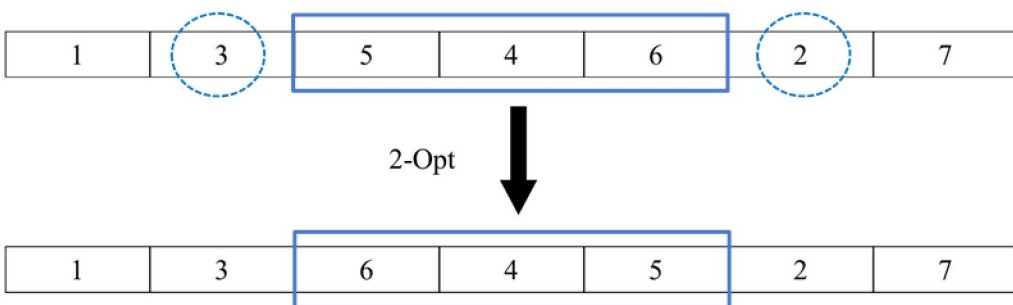

**Fig 6. An example of how to perform the 2-opt operation.**

**4.2.3 2-opt operator.** This operator is also known as 2-exchange. Randomly select two customers, and invert all the customers between them. The operation procedure is presented in Fig 6: a selected customer (framed by a dotted line) is randomly selected position, and the solid box marks the position where the inverted change occurs, and the delivery order becomes 1→3→6→4→5→2→7 after the 2-opt transformation.

In order to increase the exploration ability of the algorithm on the neighborhood, this paper defines insert, swap and 2-opt as mutation mode 1, mutation mode 2 and mutation mode 3, respectively, and more possible TSP schemes are generated by randomly selecting the types of variants to increase their diversity.

**Step 6: Tabu List**

One of the three mutation modes is randomly operated during one iteration in the neighbourhood searching process. This mechanism serves the purpose of increasing the diversity of the solutions and improving the robustness of the algorithms. Meanwhile, several recently visited solutions are forbidden for a number of iterations to prevent cycling. This function is realized by placing the solutions in a tabu list.

For the TSP/VRP problem, we set a tabu list of TGWO with a length as *length* and a width of 3. It has the ability to record the neighbourhood transform strategies and prohibit them from being performed repeatedly in a given period. The three columns of the tabu list are the code numbers for the mutation mode and two randomly chosen customers. The functional mechanism of the tabu list is shown in Fig 7, and the sample route is 1→3→5→4→6→2→7.

After generating several candidate solutions in the neighbourhood, their fitness values are stored for further comparison. If the fitness value of one mutated solution is proven to be better than that of the current solution, the TGWO algorithm verifies whether the mutation mode and corresponding customer position already exist in the tabu list. If so, this solution is abandoned and the searching process in the neighbourhood is continued. If not, the current fitter solution *better–so–far* is replaced by this one and its corresponding mutation mode is added into the tabu list, so it is not being visited again for a number of iterations.

**Step 7: Termination**

According to the maximum number of iterations set in advance, when the number of cycles of the entire calculation process of the TGWO is greater than this value, the calculation is terminated, and the optimal scheme of this algorithm and the total distance are output.

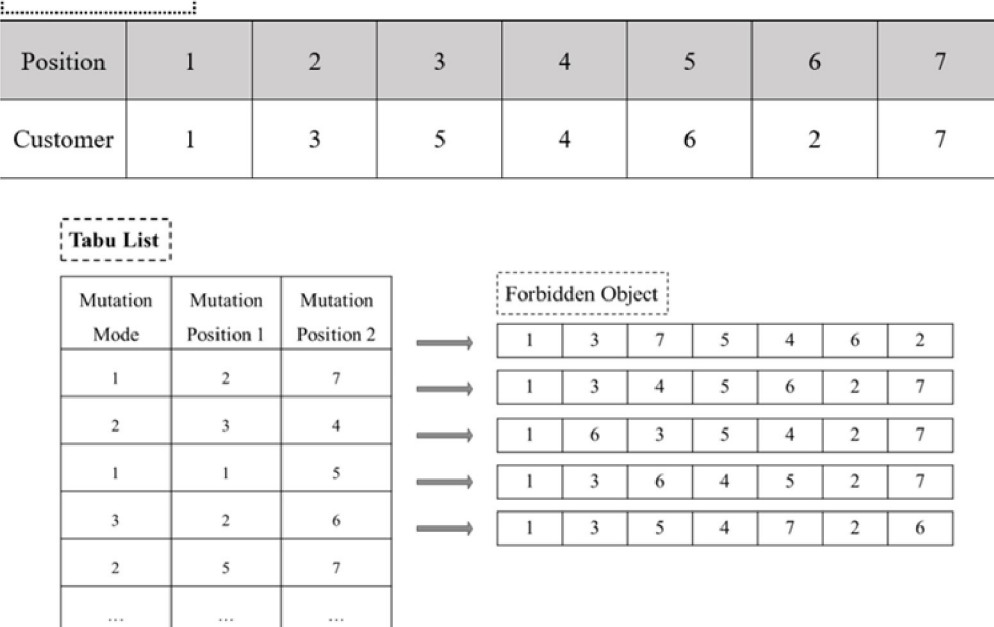

**Fig 7. Operational mechanism of the tabu list.**

## 5. Results

### 5.1 Validity of TGWO

To evaluate the TGWO algorithms proposed in this paper, TGWO is benchmarked on 6 benchmark functions. These benchmark functions are chosen from the 23 classical benchmark functions utilized by many researchers [24, 38] to compare our results to those of the current meta-heuristics. The benchmark functions used are minimization functions and divided into two groups, including three unimodal functions ($F_1$–$F_3$) and three multimodal functions ($F_4$–$F_6$). These benchmark functions are listed in Table 2, where *Dim* indicates the dimension of the function, *Range* is the boundary of the function's search space, and $f_{min}$ is the theoretical optimum. Fig 8 illustrates the 2-D versions of the unimodal and multimodal benchmark functions used. To verify the results, the TGWO algorithm is compared with the PSO, WOA and the original GWO algorithm for verification. Taking F1 as an example, the convergence curves of each algorithm in two different dimensions are shown in Fig 9. And PSO and WOA are designed according to the paper of [39] and [40], except that the fitness function is the same as that of GWO and TGWO. The lower the fitness function value is, the better the result. The parameters of the algorithms are set according to Table 3. In order to reduce the influence of randomness on the algorithm comparison results, all algorithms are run 200 times on each function for 10 and 30 dimensions and are tested in MATLAB R2020a. The operating system is Windows 10.

**5.1.1 Exploitation analysis.** According to the Fig 9, in the process of grey wolf search for prey, the neighborhood search and tabu list functions added by the improved grey wolf algorithm greatly enhance the ability of the original grey wolf population to explore and optimize the local solution, and increase the diversity of the obtained solution in the calculation process, and expand the search range in the search space, so the optimization effect of the algorithm has been greatly improved, and the TGWO algorithm provides superior results compared to those of PSO, WOA and GWO. It is especially evident in the optimization test of the unimodal functions ($F_1$, $F_2$ and $F_3$). The results show the superior performance in terms of exploiting the optimum.

**Table 2. Parameters of the test function.**

| Function | Dim | Range | $f_{min}$ |
|---|---|---|---|
| $F_1 = \sum_{i=1}^{n} x_i^2$ | 10/30 | [−100,100] | 0 |
| $F_2 = \sum_{i=1}^{n} \lvert x_i \rvert + \prod_{i=1}^{n} \lvert x_i \rvert$ | 10/30 | [−10,10] | 0 |
| $F_3 = \sum_{i=1}^{n} \left( x_i^4 + random[0,1] \right)$ | 10/30 | [-1.28,1.28] | 0 |
| $F_4 = -20 e^{-\frac{1}{5}\sqrt{\frac{1}{n}\sum_{i=1}^{n} x_i^2}} - e^{\frac{1}{n}\sum_{i=1}^{n} cos(2\pi x_i)} + 20 + e$ | 10/30 | [−32,32] | 0 |
| $F_5 = \frac{\pi}{n}\left\{ 10\, sin(\pi y_1) + \sum_{i=1}^{n-1}(y_i - 1)^2 \left[1 + 10\, sin^2(\pi y_{i+1})\right] + (y_n - 1)^2 \right\} + \sum_{i=1}^{n} u(x_i, 10, 100, 4)$ <br> $y_i = 1 + \frac{x_i+1}{4}$ <br><br> $u(x_i, a, k, m) = \begin{cases} k(x_i - a)^m & x_i > a \\ 0 & -a < x_i < a \\ k(-x_i - a)^m & x_i < -a \end{cases}$ | 10/30 | [−50,50] | 0 |
| $F_6 = 0.1\left\{ sin^2(3\pi x_1) + \sum_{i=1}^{n}(x_i - 1)^2 \left[1 + sin^2(3\pi x_{i+1} + 1) + 1\right] + (x_n - 1)^2 \left[1 + sin^2(2\pi x_n)\right] \right\} + \sum_{i=1}^{n} u(x_i, 5, 100, 4)$ | 10/30 | [−50,50] | 0 |

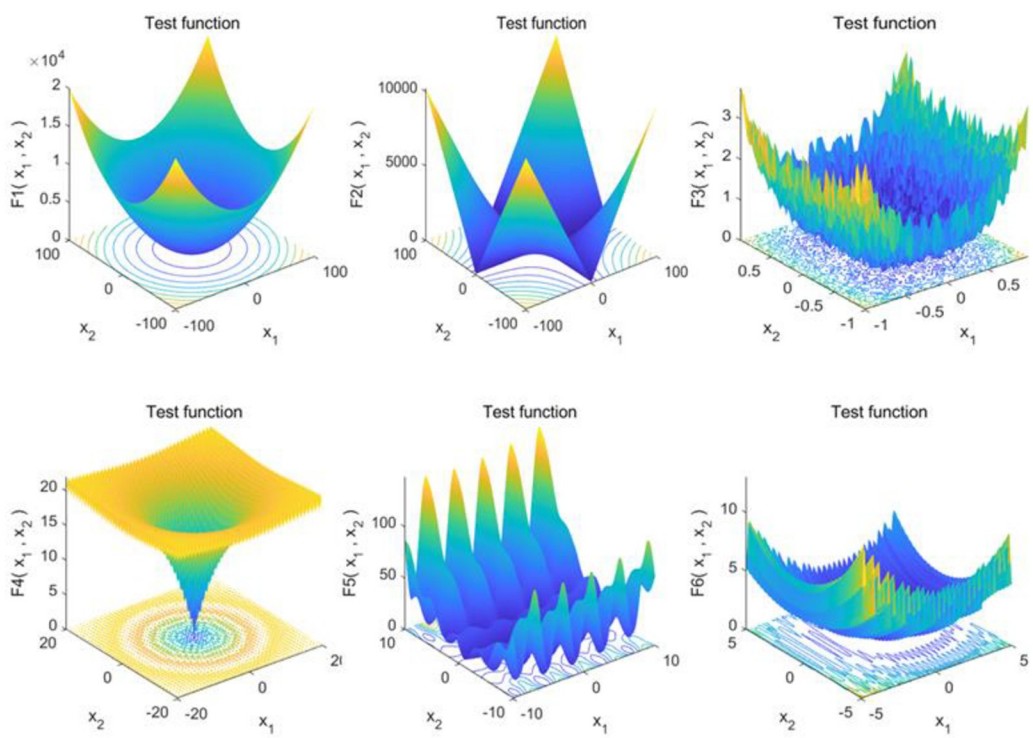

**Fig 8. 2-D versions of six benchmark functions.**

**5.1.2 Exploration analysis.** Differ from unimodal functions, multimodal functions have multiple local optima, and the number of these optima increases exponentially with the dimensionality of the function. This characteristic makes them ideal for evaluating an algorithm's exploration capabilities as a benchmark. According to the Fig 9, TGWO can provide highly competitive results on the multimodal benchmark functions. With the increase of problem dimensions, the GWO algorithm provides relatively poor convergence behaviour for exploitation. The TGWO algorithm, conversely, shows its superior performance to that of GWO. It even outperforms PSO on the results of experiments on multimodal functions $F_5$ and $F_6$. All these results show that the TGWO algorithm has merit in terms of exploration ability in search space. This also shows that the combination of the GWO algorithm and the TS algorithm is reasonable and effective, which can greatly improve the reliability of the algorithm.

## 5.2 Case study

Since fresh product e-commerce is an online trading platform, the customer's location, time window constraints and demands are constantly changing when planning the vehicle delivery route each time, so this paper selects one of the distribution centres of Company J, a fresh e-commerce platform in City B, as the research object. Besides, the distribution centre currently has 10 delivery vehicles waiting to be called, assuming that the speed of each delivery vehicle is 1, the maximum capacity is 20, and the maximum driving distance is 200, in order to complete the distribution task, each vehicle needs to start from the distribution centre 1 and finally return to it. The fixed cost of a delivery vehicle $C_0$ is 150 and the unit transportation cost is 10. (Other parameters setting: M = 100000000). In this section, the GWO, TS and TGWO are used to solve the above VRPTW problem. Among them, the solution software is MATLAB

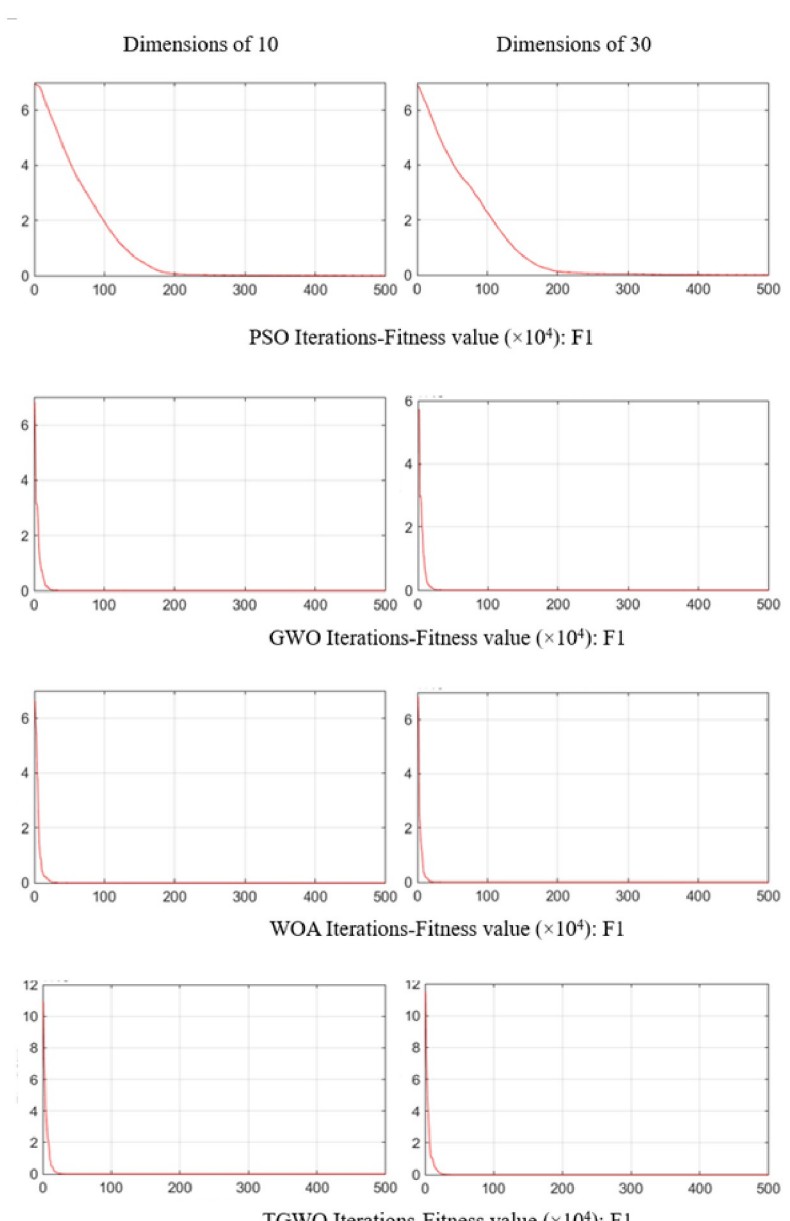

**Fig 9. Convergence curves.**

R2020a, the number of iterations is 800, the number of wolves is 80, the lower and upper bounds are set to 0.01 and 10 respectively (the upper and lower bounds of the first dimension are 0.001), the tabu list length is 10, the neighborhood search number is 80, and the tabu list length in TGWO is 30, the neighborhood search is 50 times, and the neighborhood iteration is 100 times. Finally, the optimal costs calculated by the three algorithms are 17314.8, 7754.9 and 7112.5, respectively.

The iteration processes of three algorithm are shown in Fig 10.

The VRP route obtained by the three algorithms is as follows.

**Table 3. Initial parameters of algorithms.**

| Algorithm | Parameter | Value |
|---|---|---|
| GWO | Population size ($N$) | 30 |
| | $\vec{\alpha}$ | Linearly decreased from 2 to 0 |
| | Maximum number of iterations ($M_{iter}$) | 500 |
| TGWO | Population size ($N$) | 30 |
| | $\vec{\alpha}$ | Linearly decreased from 2 to 0 |
| | Tabu length ($l$) | 10 |
| | Neighbourhood candidates ($N_{neighbor}$) | 30 |
| | Iteration of neighbourhood search ($Iter_{neighbor}$) | 50 |
| | Maximum number of iterations ($M_{iter}$) | 500 |
| PSO | Population size ($N$) | 30 |
| | $c_1$, $c_2$ | 1.4945, 1.4945 |
| | Maximum velocity of particles ($V_{max}$) | 6 |
| | Inertia weight ($w$) | Linearly decreased from 0.9 to 0.4 |
| | Maximum number of iterations ($M_{iter}$) | 500 |
| WOA | Population size ($N$) | 30 |
| | $\vec{\alpha}$ | Linearly decreased from 2 to 0 |
| | Maximum number of iterations ($M_{iter}$) | 500 |

*TS.* As shown in first figure of Fig 11, a total of 4 delivery vehicles are required to provide services to the customers in this example. Among them, the first car has a route of 1→12→14→4→16→1; the second 1→15→5→7→11→6→2→1; the third 1→10→3→9→8→13→17→1, and the fourth vehicle has a route of 1→18→1.

*GWO.* As shown in second figure of Fig 11, a total of 4 delivery vehicles are required to provide services to the customers in this example. Among them, the first car has a route of 1→10→8→17→7→9→1; the second 1→4→3→6→14→2→1; the third 1→13→11→12→5→18→1, and the fourth vehicle has a route of 1→15→16→1.

*TGWO.* As shown in last figure of Fig 11, a total of 4 delivery vehicles are required to provide services to the customers in this example. Among them, the first car has a route of 1→8→15→18→9→2→16→1; the second 1→10→11→13→7→17→1; the third 1→4→3→14→12→5→1, and the fourth vehicle has a route of 1→6→1.

The TGWO algorithm enhances the variety of available solutions by utilizing neighborhood search and tabu list functions in the solution process. This results in a more efficient and improved VRP scheme compared to the other two algorithms, with a reduced total driving distance and cost. Specifically, the TGWO algorithm saved 50.34% and 30.66% in terms of total travel distance compared to TS and GWO algorithms, respectively. In addition, it saved 143.44% and 9.03% in terms of total distribution cost, respectively. Furthermore, the VRP scheme obtained by the TGWO algorithm is clearer and more concise than the schemes obtained by TS and GWO. It arranges only one delivery person to provide services to customers in one direction as much as possible, resulting in reduced total driving distance and higher rationality. In conclusion, this paper presents a superior set of vehicle route planning schemes for the cold chain distribution process of fresh e-commerce. Comparison with the schemes obtained by different algorithms reveals that the TGWO algorithm provides a more rational, cost-effective and faster solution to the problem of cold chain distribution for fresh products. Thus, the improved grey wolf optimizer algorithm TGWO demonstrates its efficacy in solving discrete problems.

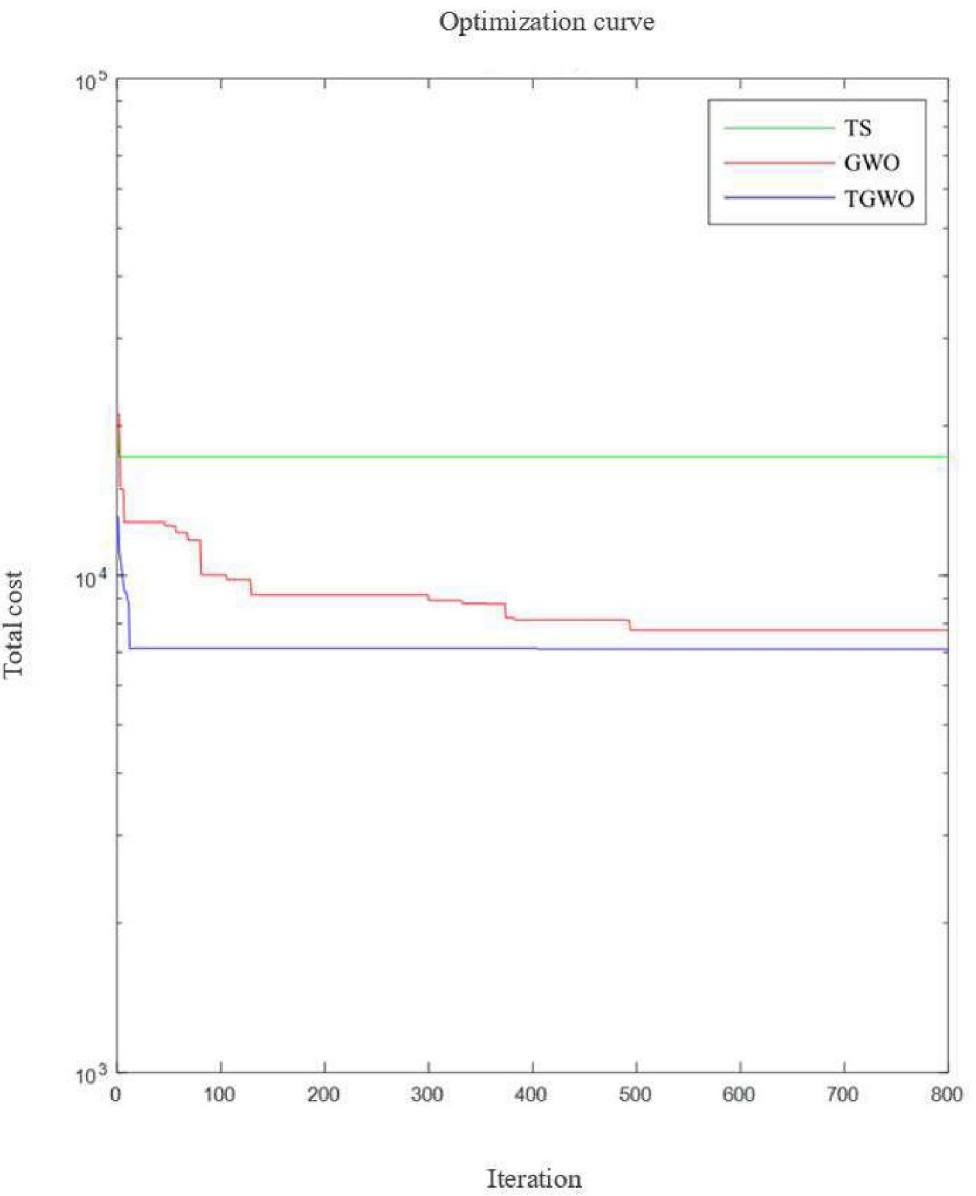

**Fig 10. Convergence graph of TS, GWO and TGWO.**

## 6. Discussion

The cold chain logistics distribution of fresh products plays a critical role in ensuring that the quality and safety of perishable goods are maintained during transportation and storage. However, optimizing the cold chain logistics distribution process can be a challenging task due to the complexity of the supply chain network, the varying demands and constraints, and the dynamic nature of the market. To address these challenges, recent studies focused more on various optimization techniques and have shown that hybridizing different optimization algorithms can lead to better performance and faster convergence. Therefore, this paper proposed a hybrid tabu-grey wolf optimizer algorithm for analyzing the cold chain distribution of fresh e-commerce and establishing a vehicle route optimization model to minimize the total

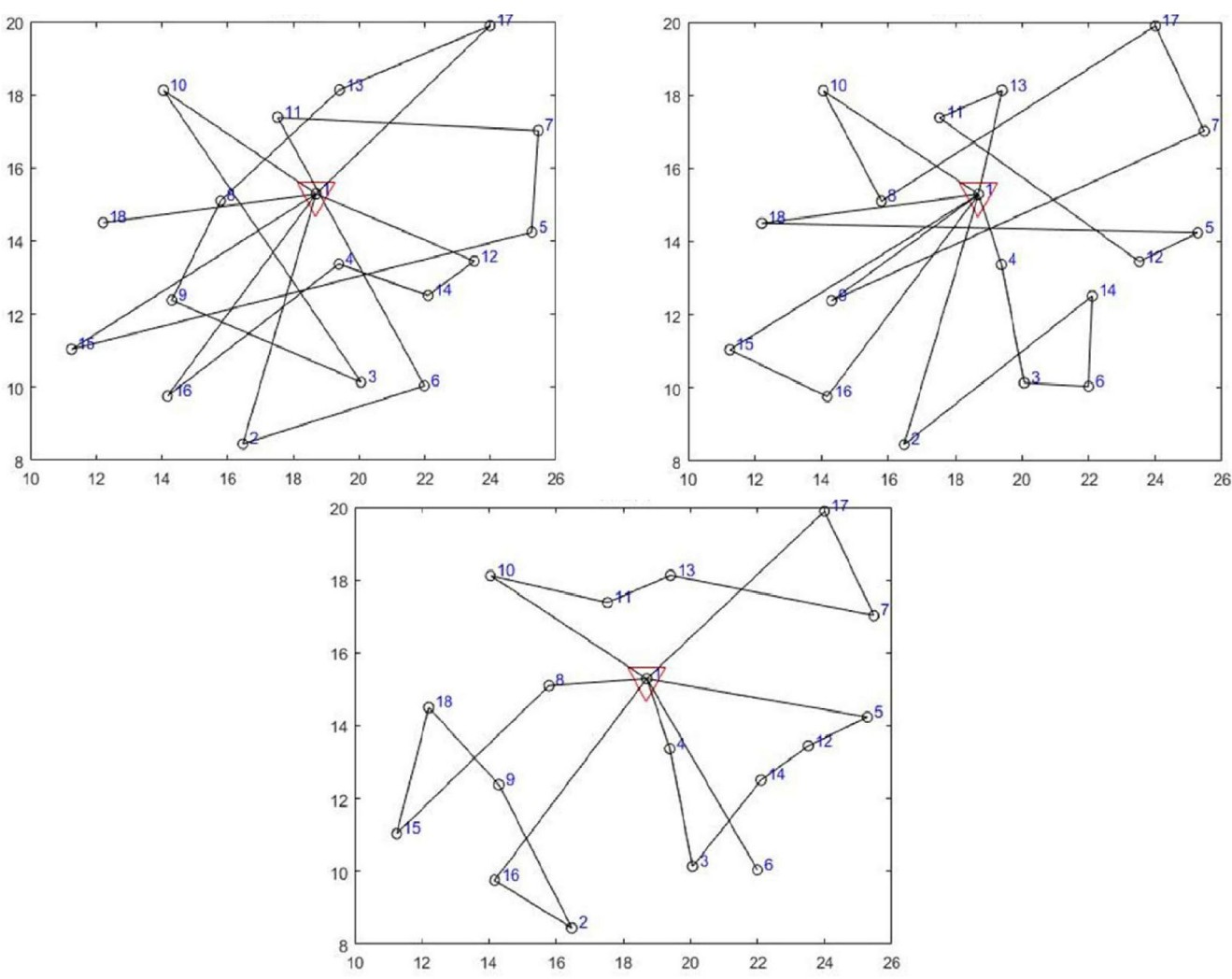

**Fig 11. The VRP solution found by TS, GWO, and TGWO.**

distribution cost of the vehicle scheduling scheme. The improved algorithm integrates Tabu Search and Grey wolf optimizer algorithms, named Tabu-Grey wolf optimizer (TGWO) algorithm, which considers multiple constraints such as vehicle capacity, maximum driving distance, and coexistence of soft and hard time windows.

## 6.1 Theoretical implications

This study makes two aspects of theoretical contributions. Firstly, by proposing a hybrid TGWO algorithm for optimizing cold chain logistics distribution. By combining the two algorithms, the TGWO algorithm offers enhanced performance and faster convergence. This finding emphasizes the significance of employing hybrid optimization techniques to tackle the challenges posed by the complex supply chain network, dynamic market dynamics, and diverse demands and constraints. Secondly, the study specifically examines the cold chain distribution of fresh products within the e-commerce domain. Through the establishment of a vehicle route optimization model, the proposed TGWO algorithm aims to minimize the total

distribution cost while considering various constraints, such as vehicle capacity, maximum driving distance, and soft and hard time windows. The effectiveness of the TGWO algorithm is demonstrated through benchmark tests and comparisons with classical optimization algorithms, including PSO, WOA and GWO. It validates the superior performance of TGWO in optimizing cold chain distribution routes for fresh products using actual distribution data from a fresh e-commerce platform. The findings contribute to the theoretical understanding of hybrid optimization algorithms in addressing the complexities of cold chain logistics distribution, particularly within the dynamic and time-sensitive environment of fresh e-commerce.

## 6.2 Managerial implications

The findings of this study have several managerial implications. The proposed hybrid TGWO offers a practical solution for optimizing the cold chain logistics distribution of fresh products. This approach enables managers to minimize the total distribution cost of vehicle scheduling schemes. The findings of this study highlight the importance of employing advanced optimization techniques to enhance the efficiency and effectiveness of cold chain logistics in preserving the quality and safety of perishable goods. Additionally, the study compares the performance of the TGWO algorithm with some classical optimization algorithms and demonstrate the superior performance of the TGWO algorithm in optimizing cold chain distribution routes for fresh products. This comparative analysis provides valuable insights for managers in selecting the most suitable algorithm for their specific distribution challenges. Managers can consider adopting the TGWO algorithm as a decision support tool to improve their cold chain logistics operations, reduce costs, and enhance overall supply chain performance.

## 6.3 Limitations and future research

This study acknowledges certain limitations that should be taken into account. This study focuses on the cold chain logistics distribution of fresh products, it is important to acknowledge the potential limitations when applying the findings to other industries or different types of products. Each industry and product category may have unique characteristics, constraints, and operational requirements that can significantly impact the effectiveness and feasibility of the proposed algorithm. Therefore, managers should exercise caution and adapt the algorithm accordingly to ensure its suitability and effectiveness in different operational settings. Additionally, the study is based on certain assumptions and constraints, which should be taken into account when applying the research findings to real-world scenarios. These assumptions may limit the generalizability of the results and require careful consideration of their applicability in specific practical situations. Managers should assess the compatibility of these assumptions and constraints with their own operational contexts to ensure the feasibility of implementing the proposed algorithm.

This study also needs some improvement in future research, such as introducing new search operators to enhance the algorithm's performance, like inter-route search operators, or route destruction and repair operators, and other novel meta-heuristic algorithms can be designed; developing new parameters, for example, fuel consumption and vehicle speed can be considered to reduce carbon emissions, or the objective of maximizing customer satisfaction can be included; comparison with other algorithms for VRPTW, such as the improved ant colony algorithm [41] or the chaotic genetic algorithm with variable neighborhood search [42], to determine the best-performing algorithm for optimizing the cold chain logistics distribution of fresh products; application to more real-world scenarios, further research can be conducted to apply the algorithm to real-world scenarios and test its effectiveness in optimizing the cold chain logistics distribution of fresh products in practice. Overall, the research on optimizing

the cold chain logistics distribution of fresh products based on a hybrid Tabu-Grey wolf optimizer algorithm has great potential for further development and improvement in the future.

## Supporting information

**S1 Data set.**
(XLSX)

## Author Contributions

**Conceptualization:** Hao Zhang.

**Visualization:** Liling Wang.

**Writing – original draft:** Liling Wang.

**Writing – review & editing:** Jianing Yan.

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
