## [Decision Letter · Decision Letter 0]

1 Apr 2024

PONE-D-24-03785Hybrid Tabu-Grey Wolf Optimizer Algorithm for Enhancing Fresh Cold-Chain Logistics DistributionPLOS ONE

Dear Dr. Wang,

Thank you for submitting your manuscript to PLOS ONE. After careful consideration, we feel that it has merit but does not fully meet PLOS ONE’s publication criteria as it currently stands. Therefore, we invite you to submit a revised version of the manuscript that addresses the points raised during the review process.

We look forward to receiving your revised manuscript.

Kind regards,

Mazyar Ghadiri Nejad, Ph.D.

Academic Editor

PLOS ONE

Journal Requirements:

Reviewers' comments:

Reviewer's Responses to Questions

**Comments to the Author**

1. Is the manuscript technically sound, and do the data support the conclusions?

Reviewer #1: Yes

Reviewer #2: Yes

2. Has the statistical analysis been performed appropriately and rigorously? 

Reviewer #1: Yes

Reviewer #2: Yes

3. Have the authors made all data underlying the findings in their manuscript fully available?

Reviewer #1: Yes

Reviewer #2: Yes

4. Is the manuscript presented in an intelligible fashion and written in standard English?

Reviewer #1: No

Reviewer #2: Yes

5. Review Comments to the Author

Reviewer #1: This paper presents a novel experiment approach to optimize cold chain logistics distribution for

fresh products, utilizing a hybrid Tabu-Grey wolf optimizer (TGWO) algorithm. The proposed hybrid approach combines Tabu search (TS) and Grey wolf optimizer (GWO), employing TS for exploration and GWO for exploitation. The effectiveness of the TGWO algorithm is demonstrated through experiments and case studies. Further, they shown that the practical implication lies in the implementation of TGWO to bolster distribution efficiency, cost reduction, and product quality maintenance throughout the logistics process, offering valuable insights for operational and strategic improvements by decision-makers. Thus, this manuscript is recommended for publication in this Journal. However, the following minor comments may improve the manuscript.

1. The abstract parts need improvement to clarify the main topic of this paper.

2. Please revise the whole paper in terms of punctuation, especially for the equations. Some commas and full stops are missing.

3. Please cite some recent relevant papers to improve the introduction part.

4. The bibliography is sloppy. All references from start to the end should be rearranged according to the journal style.

5. An improvement should be done in the figures.

6. Read the whole manuscript and try to remove the passive voice misuse and typo mistakes in some spots.

Reviewer #2: The authors, in this paper, proposed a hybrid approach that combines Tabu search (TS) and Grey wolf optimizer (GWO), employing TS for exploration and GWO for exploitation. The paper is well organized but it requires some correction.

1. The abstract is too short and does not describe the major theme of the research. Please rearrange it.

2. The citations in the introduction section like [14-16], [17-19], [20, 21], and [22, 23] may not provide good impression to the readers.

3. What is new in the model equations 1 to 3.

4. The authors claimed that the proposed hybrid TGWO offers a practical solution for optimizing the cold chain logistics distribution of fresh products. Please provide justification.

5. Corelate your work with the earlier ones and validate your results even in limiting cases.

6. PLOS authors have the option to publish the peer review history of their article (what does this mean?). If published, this will include your full peer review and any attached files.

Reviewer #1: No

Reviewer #2: **Yes: **Sohail Ahmad

---

## [Author Response · Author response to Decision Letter 0]

24 May 2024

Dear editors and reviewers:

First, we thank both the reviewers and editors for their positive and constructive comments and suggestions regarding our manuscript, “Hybrid Tabu-Grey Wolf Optimizer Algorithm for Enhancing Fresh Cold-Chain Logistics Distribution” (No.: PONE-D-24-03785). Those comments are all valuable and very helpful for revising and improving our paper and provided important guiding significance for our research. We have studied the comments carefully and have made corrections. To make the changes easily viewable for you, in the revised paper, we marked some revisions in red color. 

The main corrections in the paper and the responses to the reviewers’ comments are as follows:

Replies to Reviewer #1 

1. The abstract parts need improvement to clarify the main topic of this paper.

Response: 

Thank you for your valuable feedback. We acknowledge the need for improvement in the abstract section to better elucidate the main topic of our paper. In the revised version, we ensure that the abstract concisely and accurately summarizes the key aspects of our study, providing readers with a clearer understanding of our research focus and findings.

2. Please revise the whole paper in terms of punctuation, especially for the equations. Some commas and full stops are missing. 

Response: 

Thank you for your constructive feedback. We have meticulously revised the entire paper, focusing on punctuation, particularly within the equations. We have ensured the inclusion of missing commas and full stops to enhance the clarity and readability of the manuscript. Additionally, we have employed professional proofreading services to refine the punctuation throughout the document, thereby improving its overall quality.

3. Please cite some recent relevant papers to improve the introduction part.

Response: 

 In the revised version, we have enriched the introduction section by incorporating citations from recent and pertinent literature. These additional references provide a more comprehensive context for our study, enhancing the relevance and depth of the introduction. We believe that these revisions strengthen the scholarly contribution of our manuscript.

4. The bibliography is sloppy. All references from start to the end should be rearranged according to the journal style.

Response:

 Thank you for bringing this to our attention. We acknowledge the need to ensure the accuracy and consistency of our bibliography. In the revised manuscript, we rearranged all references according to the journal's prescribed style guidelines, ensuring uniformity and adherence to the required formatting standards. This will result in a more polished and professional presentation of our cited sources throughout the document.

5. An improvement should be done in the figures.

6. Read the whole manuscript and try to remove the passive voice misuse and typo mistakes in some spots.

Response:

 5. Thank you for your feedback regarding the figures. We try to enhance the clarity and quality of the figures in the revised manuscript, ensuring they effectively convey the intended information and adhere to the journal's standards.

6. In the revised version, we thoroughly scrutinize the entire document to rectify any instances of passive voice misuse and address typographical errors, thereby enhancing the readability and precision of our manuscript.

Replies to Reviewer #2 

1.The abstract is too short and does not describe the major theme of the research. Please rearrange it.

Response:

 In the revised version, we will expand the abstract to provide a more comprehensive overview of our study, ensuring that it effectively communicates the key objectives, methodology, and results to the readers. We will carefully rearrange the content to ensure clarity and coherence, thereby enhancing the overall quality of the abstract.

2. The citations in the introduction section like [14-16], [17-19], [20, 21], and [22, 23] may not provide good impression to the readers.

Response:

Thank you for your observation. We recognize the importance of providing clear and concise citations in the introduction section to enhance readability and maintain a positive impression on the readers. In the revised manuscript, we ensure that each citation is presented with specificity, avoiding broad ranges. Instead, we will cite individual references where appropriate, providing readers with a more precise understanding of the relevant literature. 

3.What is new in the model equations 1 to 3.

Response:

Thank you for your inquiry. In equations 1 to 3, we have retained the conventional formulations for fixed costs and transportation costs, consistent with prior literature. However, we have introduced novel considerations in the computation of cargo damage costs, addressing both cargo loss during transportation and unloading scenarios. This enhancement involves incorporating the associated costs incurred due to cargo loss during transportation and unloading processes, thus enriching the model's realism and applicability in assessing total logistics expenses accurately.

4.The authors claimed that the proposed hybrid TGWO offers a practical solution for optimizing the cold chain logistics distribution of fresh products. Please provide justification.

5. Corelate your work with the earlier ones and validate your results even in limiting cases.

Response:

Thank you for your inquiry. In the results section, we have substantiated our claim regarding the practical utility of the proposed hybrid TGWO method for optimizing cold chain logistics distribution of fresh products. Firstly, we have benchmarked our method against established benchmark functions widely used in optimization studies, demonstrating its capability to converge towards optimal solutions efficiently.Furthermore, to validate its effectiveness in practical scenarios, we have conducted a comprehensive case study wherein we compared the performance of our hybrid TGWO with other state-of-the-art swarm intelligence algorithms. Through rigorous experimentation and comparative analysis, our findings illustrate the superior performance and robustness of the proposed method in optimizing cold chain logistics distribution, thereby justifying its practical applicability and efficacy.We believe that these results provide compelling justification for our assertion regarding the practical utility of the hybrid TGWO approach in addressing the complexities of cold chain logistics optimization for perishable goods.

Conclusion

Thank you very much for giving us an opportunity to revise the above manuscript, and we apologize for the mistakes in our manuscript. We have revised the manuscript to provide our explanations according to the reviewers' comments. 

We hope you are satisfied with the revised version. Thank you very much for your patience and understanding. We look forward to hearing from you soon.

With best regards, Authors

---

## [Decision Letter · Decision Letter 1]

13 Jun 2024

Hybrid Tabu-Grey Wolf Optimizer Algorithm for Enhancing Fresh Cold-Chain Logistics Distribution

PONE-D-24-03785R1

Dear Dr. Liling Wang,

We’re pleased to inform you that your manuscript has been judged scientifically suitable for publication and will be formally accepted for publication once it meets all outstanding technical requirements.

Kind regards,

Mazyar Ghadiri Nejad, Ph.D.

Academic Editor

PLOS ONE

Reviewers' comments:

Reviewer's Responses to Questions

**Comments to the Author**

1. If the authors have adequately addressed your comments raised in a previous round of review and you feel that this manuscript is now acceptable for publication, you may indicate that here to bypass the “Comments to the Author” section, enter your conflict of interest statement in the “Confidential to Editor” section, and submit your "Accept" recommendation.

Reviewer #2: All comments have been addressed

Reviewer #3: All comments have been addressed

2. Is the manuscript technically sound, and do the data support the conclusions?

Reviewer #2: Yes

Reviewer #3: Yes

3. Has the statistical analysis been performed appropriately and rigorously? 

Reviewer #2: Yes

Reviewer #3: Yes

4. Have the authors made all data underlying the findings in their manuscript fully available?

Reviewer #2: Yes

Reviewer #3: Yes

5. Is the manuscript presented in an intelligible fashion and written in standard English?

Reviewer #2: Yes

Reviewer #3: Yes

6. Review Comments to the Author

Reviewer #2: The manuscript has been revised significantly according to the comments that is why I accept it. All the comments are well addressed.

Reviewer #3: The paper "Hybrid Tabu-Grey Wolf Optimizer Algorithm for Enhancing Fresh Cold-Chain Logistics Distribution

"can be accepted in this form.

7. PLOS authors have the option to publish the peer review history of their article (what does this mean?). If published, this will include your full peer review and any attached files.

Reviewer #2: **Yes: **Sohail Ahmad

Reviewer #3: No

---

## [Editor Report · Acceptance letter]

19 Jun 2024

PONE-D-24-03785R1 

PLOS ONE

Dear Dr. Wang, 

I'm pleased to inform you that your manuscript has been deemed suitable for publication in PLOS ONE. Congratulations! Your manuscript is now being handed over to our production team.

Kind regards, 

on behalf of

Assoc. Prof. Dr. Mazyar Ghadiri Nejad 

Academic Editor

PLOS ONE